# One Graph can Generalize: Graph-guided Structural Transfer for Source-free Open-set Domain-Adaptive Object Detection

## Abstract

Domain Adaptive Object Detection (DAOD) transfers detection capabilities from a labeled source domain to an unlabeled target domain with different visual characteristics and distributions. However, current DAOD methods often assume closed sets and require access to both source and target data, limiting their practical deployment. To address these challenges, we reformulate DAOD by explicitly considering three shifts: domain distribution shift (*Shift i*), open-set class shift (*Shift ii*), and source-free transfer shift (*Shift iii*). We propose GraphGen, a unified graph-based framework that models structural relationships between objects, enabling knowledge transfer without source data via dynamically updated graphs. The framework integrates modules for graph-based feature alignment, novelty discovery, and self-regularization. Experiments on benchmarks show that GraphGen outperforms state-of-the-art methods, with notable improvements in novel class discovery while maintaining strong performance on known categories.

## 1 Introduction

Domain Adaptive Object Detection (DAOD) aims to transfer detection capabilities from a labeled source domain to an unlabeled target domain with different visual characteristics Chen et al. (2018); Saito et al. (2019). This task is crucial for real-world deployment of detection systems, where variations in lighting, weather conditions, and imaging sensors inevitably cause visual distribution shifts between training and operational environments. Traditional DAOD approaches typically focus on aligning feature distributions between domains through adversarial training and domain-invariant representation learning Xu et al. (2020); Chen et al. (2021b).

However, existing DAOD methods face significant limitations for practical use. Most approaches assume object categories are identical across domains Wang et al. (2019); Chen et al. (2020), making them unable to handle novel categories in deployment. Conventional methods also require access to both source and target data during adaptation Li et al. (2020), which is often infeasible in industrial settings. Current techniques struggle to distinguish between domain-shifted known objects and genuinely novel categories Sindagi et al. (2020).

To address these challenges, we revisit the DAOD problem by explicitly considering three types of shifts in real-world object detection. First, domain distribution shift (*Shift i*) occurs when visual characteristics differ between source and target domains due to factors like lighting or sensor types Wang et al. (2019); Chen et al. (2020). Second, open-set class shift (*Shift ii*) arises when the target domain contains object categories absent in the source Joseph et al. (2021); Miller et al. (2021). Third, source-free transfer shift (*Shift iii*) restricts access to source training data during adaptation Kundu et al. (2020); Liang et al. (2020).

To address these challenges, we propose GraphGen, a unified graph-based framework that simultaneously tackles all three shifts. Our approach models structural relationships between objects, enabling knowledge transfer without source data access through dynamically updated graphs that capture cross-domain similarities. The framework integrates modules for feature alignment, novelty discovery, and self-regularization to address source-free open-set domain adaptation challenges. Importantly, the three modules are designed to be compatible: they act on different signals (alignment statistics, novelty scores, teacher-student consistency) but share the same instance graph,

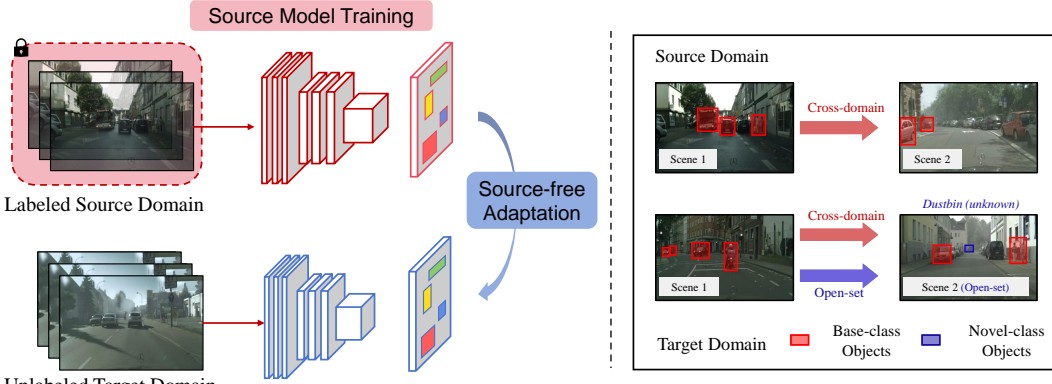

Figure 1: Overview of GraphGen for source-free domain-adaptive open-set object detection. Top: Source-free adaptation from labeled source to unlabeled target domain, highlighting base and novel class objects. Bottom: Structural graph reasoning captures cross-domain relationships for improved novel class detection.

with their objectives reinforcing each other rather than conflicting—for example, novelty discovery is uncertainty-aware so that ambiguous proposals contribute less to prototype updates, reducing conflict with alignment and self-regularization. Extensive experiments on Cityscapes→Foggy Cityscapes and Pascal VOC→Clipart benchmarks demonstrate that GraphGen outperforms existing methods in both known and novel class detection. Our key contributions include:

- A unified graph-based framework addressing domain distribution shift (**Shift i**), open-set class shift (**Shift ii**), and source-free transfer shift (**Shift iii**), enabling adaptation to unknown classes while maintaining strong performance for known categories.

- A novel graph architecture with three components: cross-domain graph alignment, graph-guided novelty discovery, and graph-aware self-regularization, each tackling one of the three key shifts.

- Comprehensive evaluation across various domain shift scenarios with unknown classes, demonstrating significant improvements over state-of-the-art methods in both detection accuracy and novel class discovery.

## 2 RELATED WORK

**Unsupervised Domain Adaptation for Object Detection.** Adapting object detectors across domains transfers capabilities from annotated source data to target scenarios with distribution shifts Chen et al. (2018); Xu et al. (2020). Recent research has developed frameworks such as adversarial techniques Chen et al. (2018); Zhu et al. (2019) with discriminator networks, self-training with confidence-based pseudo-labeling Kim et al. (2019); Rezaeianaran et al. (2021), and distribution alignment Saito et al. (2019). However, a key limitation is their reliance on closed-set assumptions, where object categories remain unchanged across domains Wang et al. (2019); Chen et al. (2020). This is impractical in real-world scenarios where target environments often contain unseen object types, reducing adaptation performance Saito et al. (2020); Fu et al. (2020). Our work addresses these issues by developing a unified framework that combines open-set recognition with domain adaptation.

**Source-Free Domain Adaptation for Object Detection.** Source-free domain adaptation tackles scenarios where source data is inaccessible during adaptation due to privacy or proprietary concerns. Early SFDA research focused on classification tasks Sindagi et al. (2020); Sakaridis et al. (2018) before expanding to object detection. Existing SFDA methods include pseudo-label self-training Li et al. (2021); VS et al. (2023), knowledge distillation Hegde et al. (2021), self-supervised feature learning Huang et al. (2021), and prototype-based adaptation Zhang et al. (2021). Recent works also explore memory banks Wang et al. (2023) and multi-teacher knowledge transfer Liu et al. (2021). While some studies have used graph-based methods for traditional domain adaptation Chen et al. (2021a), they usually require source data. Our work advances SFDA by introducing a bipartite graph

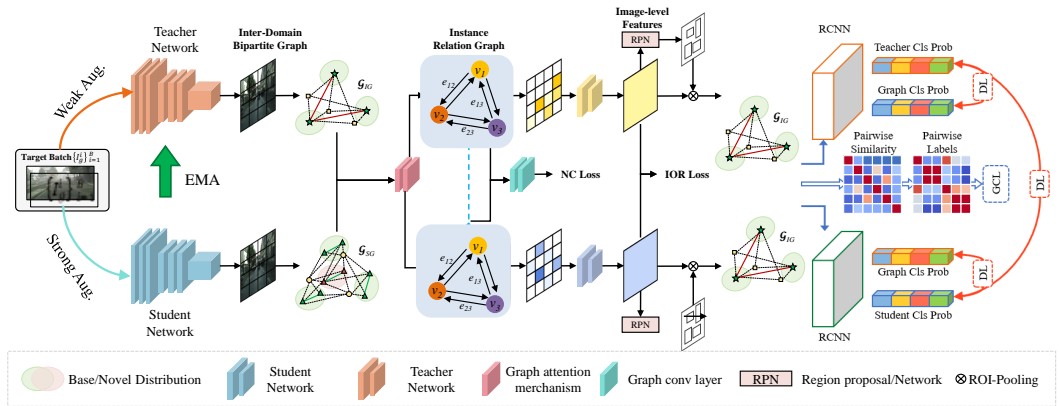

Figure 2: Overview of GraphGen for source-free domain-adaptive open-set object detection. The framework comprises three components: **(i)** cross-domain graph alignment builds an instance-level graph over target proposals and aligns features with graph convolution and cyclic consistency; **(ii)** graph-guided novelty discovery separates known and unknown objects using prototype-based scoring with normalized entropy; **(iii)** graph-aware self-regularization enforces teacher–student consistency on graph features under asymmetric augmentations in the source-free setting.

framework that models cross-domain interactions dynamically, enabling adaptation without source data.

**Object Detection in Open-world Environments.** Open-world object detection extends beyond traditional paradigms by handling both known and novel object categories, addressing limitations of frameworks like Faster R-CNN Ren et al. (2015) and YOLO Redmon et al. (2016). Recent methods (e.g., OpenDet Han et al. (2022)) engineer feature spaces or energy-based scores to better represent unknowns, and generative or distance-based approaches Liu et al. (2020); Bendale & Boult (2016) further improve robustness. Our work instead proposes a graph-based architecture that integrates structural relationship modeling with robust feature learning to detect unknown objects while maintaining performance on known categories. Unlike open-vocabulary or vision-language detectors that rely on large-scale vision–language pretraining and a text encoder over an open label space, GraphGen operates under a fixed known-class vocabulary and a strict source-free domain-adaptation protocol, making it complementary to open-vocabulary detectors.

**Graph-based and Uncertainty-aware Open-set DA/DAOD.** Existing graph-based and uncertainty-based open-set DA methods usually assume source-available settings or focus on classification. In contrast, GraphGen addresses *source-free, open-set, object detection*: source images are discarded after pre-training, and we operate on instance-level graphs built purely on target proposals with domain-agnostic prototypes. A single graph supports feature alignment, novelty discovery, and self-regularization, while entropy-aware uncertainty both detects unknowns and weights node contributions, tightly coupling graph structure, prototypes, and uncertainty under the source-free constraint.

## 3 PROPOSED METHOD

**Problem Formulation and Motivation.** In real-world object detection, three main distribution shifts must be addressed. First, *Shift i* is the domain distribution shift: given a source dataset $\mathcal{D}_s = \{(I_s^i, Y_s^i)\}_{i=1}^{n_s}$ and a target dataset $\mathcal{D}_t = \{I_t^j\}_{j=1}^{n_t}$, their distributions differ, i.e., $\mathcal{P}(I_s) \neq \mathcal{P}(I_t)$. Second, *Shift ii* is the open-set class shift: the target label set includes both known classes from the source ($\Omega_b = \{1, 2, \ldots, K\}$) and unknown categories absent in the source, grouped as a new class ($K + 1$). Third, *Shift iii* is the source-free transfer shift (Source-Free Domain Adaptation, SFDA): source data is inaccessible during adaptation, and only a pre-trained source model $\Theta_s$ is available. Crucially, after supervised training on the labeled source domain, all source images are discarded; adaptation uses only unlabeled target data $\mathcal{D}_t$.

Our goal is to adapt $\Theta_s$ to $\mathcal{D}_t$ for accurate recognition of known classes $\Omega_b$ and effective detection of unknown objects $\Omega_u$. We propose a unified framework to address all three shifts. The following sections describe each component.

**Cross-domain Graph Alignment over *Shift i*.** To address the domain shift between source and target distributions, we leverage the pre-trained model $\Theta_s$ and unlabeled target data $\mathcal{D}_t$ without access to source data $\mathcal{D}_s$. Our approach exploits the observation that while visual appearances change across domains, structural relationships between objects often remain relatively consistent - for instance, cars tend to maintain similar spatial relationships to pedestrians regardless of conditions.

We extract object query embeddings $\{\mathbf{z}_i \in \mathbb{R}^d\}_{i=1}^N$ from target samples $I_t^j \in \mathcal{D}_t$ using the feature extractor $\mathcal{F}$ from $\Theta_s$. Graph nodes are object queries (proposals) whose classification scores exceed a fixed confidence threshold, which removes very low-confidence proposals while preserving all reasonably likely objects. We construct graphs at the *mini-batch* level: each graph contains the filtered proposals from all images in the batch, yielding typically a few hundred nodes per batch. These embeddings form vertices in a graph $\mathcal{G} = (\mathcal{V}, \mathcal{E})$ where each vertex $v_i \in \mathcal{V}$ corresponds to an embedding $\mathbf{z}_i$, explicitly modeling structural relationships between objects.

To quantify object relationships in this graph, we define edge weights using a cosine similarity-based affinity measure with an exponential function. In modern transformer-based detectors, object query embeddings integrate both appearance and spatial context through cross-attention with the backbone features, so cosine similarity between query embeddings reflects similarity in a *joint semantic–spatial* space, not purely low-level appearance. When we connect proposals with high similarity in this learned embedding space, we effectively link objects that play similar roles in a scene (e.g., cars on roads, pedestrians near crossings), capturing relational patterns that remain stable across domains.

$$\omega_{ij} = \exp\left(\gamma \cdot \frac{\langle \mathbf{z}_i, \mathbf{z}_j \rangle}{\|\mathbf{z}_i\|_2 \|\mathbf{z}_j\|_2}\right), \tag{1}$$

where $\gamma > 0$ is a scalar parameter controlling angular sensitivity. Higher $\gamma$ values create sharper distinctions between similar and dissimilar objects, helping differentiate between known and potentially unknown objects based on their relationship patterns. Pairwise affinities form matrix $\Omega = [\omega_{ij}]$ capturing relationship structure between objects. To prevent instability, we normalize using degree normalization to create a normalized adjacency matrix:

$$\widetilde{\mathbf{A}} = \mathbf{D}^{-\frac{1}{2}} \, \Omega \, \mathbf{D}^{-\frac{1}{2}}, \tag{2}$$

where $\mathbf{D}$ is a diagonal matrix with $D_{ii} = \sum_j \omega_{ij}$. This normalization ensures balanced contribution from all nodes regardless of their connectivity.

With the normalized adjacency matrix, we propagate information across the graph to incorporate contextual information from related objects, enriching feature representations with structural knowledge that helps bridge the domain gap. We implement this through a graph convolution operation:

$$\tilde{\mathbf{Z}} = \Psi\left(\widetilde{\mathbf{A}} \, \mathbf{Z} \, \mathbf{W}\right), \tag{3}$$

where $\mathbf{Z} = [\mathbf{z}_1, \ldots, \mathbf{z}_N]^\top$ is the matrix of all object embeddings, $\mathbf{W}$ is a learnable weight matrix that transforms the features into a suitable space for propagation, and $\Psi(\cdot)$ is a non-linear activation function (in our implementation, GELU) that introduces non-linearity into the propagation process. This graph convolution essentially computes weighted averages of neighboring features, with the weights determined by the normalized adjacency matrix $\widetilde{\mathbf{A}}$.

To ensure structural information enriches representations without distorting their original semantic meanings, we introduce a cyclic consistency constraint that encourages graph-processed features to maintain consistency with the original features:

$$\mathcal{L}_{\text{graph-dom}} = \frac{1}{N} \sum_{i=1}^N \left\| \mathbf{z}_i - \Xi\left(\tilde{\mathbf{z}}_i\right) \right\|_2^2, \tag{4}$$

where $\Xi(\cdot)$ is a projection operator (two-layer MLP) that maps graph-processed features back to the original space. This cyclic consistency prevents feature drift and preserves semantic content from the source domain.

By integrating graph construction, feature propagation, and cyclic consistency, our model achieves effective cross-domain alignment. This approach addresses domain shift (**Shift i**) and lays the foundation for handling open-set challenges (**Shift ii**).

**Graph-guided Novelty Discovery over *Shift ii*.** Having established a robust graph-based framework to address the domain shift (**Shift i**), we now focus on the open-set recognition problem (**Shift ii**). While our graph alignment bridges distribution gaps, it doesn't inherently distinguish between known classes ($\Omega_b$) and unknown classes appearing only in the target domain.

Our key insight is that graph structure encodes relational patterns distinguishing known from unknown objects. Known objects form consistent clusters. We leverage this by integrating a dynamic semantic memory mechanism. For each known class $k \in \Omega_b$, we maintain a prototype vector $\mathbf{p}^k$ (class centroid) and a variability descriptor $\mathbf{v}^k$ (distribution spread). To determine if embedding $\mathbf{z}_i$ is known or unknown, we define a novelty score $\wp(\mathbf{z}_i)$ based on class probability and prediction entropy:

$$\wp(\mathbf{z}_i) = 1 - \max_{c \in \Omega_b}\Big\{ p(c \mid \mathbf{z}_i)\Big[ 1 - \bar{\mathcal{H}}\big(\mathbf{p}(\mathbf{z}_i)\big)\Big]\Big\}, \tag{5}$$

where $p(c \mid \mathbf{z}_i)$ is the detector's probability for class $c$. The term $\max_{c \in \Omega_b} p(c \mid \mathbf{z}_i)$ reflects model confidence, while normalized entropy $\bar{\mathcal{H}}(\mathbf{p}(\mathbf{z}_i))$ measures uncertainty:

$$\bar{\mathcal{H}}\big(\mathbf{p}(\mathbf{z}_i)\big) = -\frac{1}{\log K} \sum_{c \in \Omega_b} p(c \mid \mathbf{z}_i) \log p(c \mid \mathbf{z}_i), \tag{6}$$

where $K$ is the number of known classes, ensuring $\bar{\mathcal{H}} \in [0,1]$ and making thresholds dataset-independent. The multiplicative term $[1 - \bar{\mathcal{H}}(\mathbf{p}(\mathbf{z}_i))]$ ensures confident predictions yield lower novelty scores, while uncertain ones yield higher scores, reducing false positives for domain-shifted known objects.

For open-set adaptation, prototypes and variability descriptors must evolve to reflect target domain shifts. We use a momentum-based update: for each class $k$,

$$\mathbf{p}^k \leftarrow \tau\, \mathbf{p}^k + (1 - \tau)\, \mathbb{E}\Big[\mathbf{z}_i \mid y_i = k\Big], \tag{7}$$

where $\mathbb{E}[\mathbf{z}_i \mid y_i = k]$ is the mean embedding for class $k$, and $\tau \in (0,1)$ controls adaptation rate. The variability descriptor is similarly updated:

$$\mathbf{v}^k \leftarrow \tau\, \mathbf{v}^k + (1 - \tau)\, \mathrm{Var}\Big[\mathbf{z}_i \mid y_i = k\Big], \tag{8}$$

where $\mathrm{Var}[\mathbf{z}_i \mid y_i = k]$ is the variance for class $k$. This dual tracking enables adaptive, nuanced decision boundaries anchored to source knowledge.

To further boost discrimination, we add contrastive learning within the graph. Unknown objects may share features with known classes but have distinct relational patterns. We define a graph-based contrastive loss to pull same-class embeddings closer and push different-class ones apart:

$$\mathcal{L}_{\text{graph-nov}} = -\frac{1}{|\mathcal{P}|} \sum_{(i,j) \in \mathcal{P}} \log \frac{\exp\Big( \langle \tilde{\mathbf{z}}_i, \tilde{\mathbf{z}}_j \rangle / \tau_c \Big)}{\sum_{k \in \mathcal{N}(i)} \exp\Big( \langle \tilde{\mathbf{z}}_i, \tilde{\mathbf{z}}_k \rangle / \tau_c \Big)}, \tag{9}$$

where $\langle \cdot, \cdot \rangle$ is the standard dot product in the feature space, $\tau_c$ controls concentration, $\mathcal{P}$ is the set of positive pairs, and $\mathcal{N}(i)$ is the set of negatives for $i$. Both known and pseudo-labeled unknowns are used. Treating all novel objects as a single "unknown" class, the loss encourages compact, distinct representations for novel instances, separating them from known classes.

This contrastive mechanism operates on graph-transformed features $\tilde{\mathbf{z}}_i$, grouping objects by structural patterns beyond visual similarity. By combining prototype-based novelty detection and graph-structured contrastive learning, our approach addresses the open-set challenge (**Shift [ii]**) with adaptive boundaries that identify novel categories while preserving known-class performance.

**Graph-aware Self-regularization over *Shift iii*.** Having addressed both the domain shift (**Shift i**) through cross-domain graph alignment and the open-set challenge (**Shift ii**) through graph-guided

novelty discovery, we now confront the third challenge: adapting to the target domain without source data access (***Shift iii***). In our source-free scenario, we are limited to using only a pre-trained source model $\Theta_s$ and unlabeled target data $\mathcal{D}_t$, creating a fundamental challenge of adapting to the target domain while preserving source knowledge.

Our solution creates a self-supervised framework that maintains consistency between different views of the same target data. Inspired by Mean-Teacher paradigm, we establish a teacher-student knowledge distillation framework uniquely adapted to our graph-based representation learning approach. For each target image $I_t \in \mathcal{D}_t$, we generate two distinct augmented versions using different stochastic transformation operators:

$$\widetilde{I}_t = \mathcal{T}_s(I_t) \quad \text{and} \quad \widehat{I}_t = \mathcal{T}_w(I_t), \tag{10}$$

where $\mathcal{T}_s(\cdot)$ represents a strong augmentation operator applying aggressive transformations (color jittering, cropping, blur), and $\mathcal{T}_w(\cdot)$ represents a weak augmentation operator applying milder transformations (horizontal flipping, slight color adjustments). This asymmetric strategy challenges the student network while maintaining stable teacher predictions.

Our framework employs two identical networks with different update mechanisms: a student network $\Theta_S$ processing strongly augmented inputs $\widetilde{I}_t$, and a teacher network $\Theta_T$ processing weakly augmented inputs $\widehat{I}_t$. The teacher's parameters are updated using an exponential moving average (EMA) of the student's parameters:

$$\Theta_T \leftarrow \alpha\,\Theta_T + (1 - \alpha)\,\Theta_S, \tag{11}$$

where $\alpha \in (0, 1)$ is a decay factor (typically 0.99) controlling adaptation rate. This EMA mechanism ensures the teacher evolves gradually, providing reliable pseudo-labels for self-supervised learning.

The key innovation in our source-free adaptation is integrating graph-based reasoning into pseudo-labeling. For each object query $q_t$ processed through both networks, we determine whether to assign it a known class label, an unknown class label, or exclude it due to uncertainty, leveraging both class probabilities and novelty scores from our graph-guided novelty discovery module:

$$\tilde{y}_t = \begin{cases} \arg\max_{c \in \Omega_b} p(c \mid q_t), & \text{if } \max_{c \in \Omega_b} p(c \mid q_t) \geq \tau_{\text{known}} \\ K + 1, & \text{if } \max_{c \in \Omega_b} p(c \mid q_t) < \tau_{\text{known}} \\ & \quad \text{and} \quad \wp(q_t) \geq \tau_{\text{unknown}} \\ \text{ignored}, & \text{otherwise} \end{cases} \tag{12}$$

where $\tau_{\text{known}}$ and $\tau_{\text{unknown}}$ are thresholds controlling pseudo-labeling strictness, preventing error accumulation and model drift in the source-free setting. The probabilities $p(c|q_t)$ are obtained from the classification head of the detector.

The graph-transformed features $\tilde{\mathbf{z}}_i$ capture contextual relationships between objects, providing a holistic classification basis. We formulate a consistency loss between teacher and student networks:

$$\mathcal{L}_{\text{graph}-\text{sf}} = \frac{1}{N} \sum_{i=1}^{N} \left\| \tilde{\mathbf{z}}_i^{(T)} - \tilde{\mathbf{z}}_i^{(S)} \right\|_2^2, \tag{13}$$

where $\tilde{\mathbf{z}}_i^{(T)}$ and $\tilde{\mathbf{z}}_i^{(S)}$ represent the graph-transformed features of the $i$-th object query from the teacher and student networks, respectively. This loss ensures that the structural relationships captured by the student network remain consistent with those of the teacher network, which is essential for preserving the source domain knowledge while adapting to the target domain.

By leveraging graph-based representations and consistency constraints, our approach effectively addresses the source-free challenge (***Shift iii***), completing our unified framework for tackling all three major shifts in domain-adaptive open-set object detection.

**Overall Optimization Objective.** After addressing each challenge, we integrate all components into a unified optimization framework. The total objective combines detection loss and our regularization terms:

$$\mathcal{L}_{\text{total}} = \mathcal{L}_{\text{det}} + \lambda_1 \mathcal{L}_{\text{graph}-\text{dom}} + \lambda_2 \mathcal{L}_{\text{graph}-\text{nov}} + \lambda_3 \mathcal{L}_{\text{graph}-\text{sf}}. \tag{14}$$

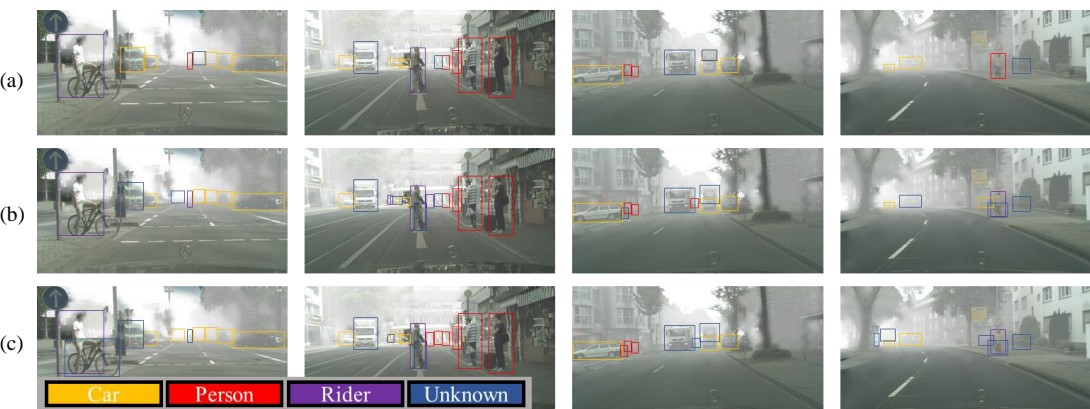

Figure 3: Qualitative comparison on AOOD scenarios from Cityscapes to Foggy among (a) DDETR [60], (b) OW-DETR [18], (c) Ours.

Table 1: Quantitative comparison on source-free open-set object detection from Cityscapes to Foggy Cityscapes. GraphGen consistently outperforms baselines across all metrics, with **Top-1** and Top-2 performance marked.

| Method | Set | num. novel-class: 3 | | | | num. novel-class: 4 | | | | num. novel-class: 5 | | | |
|---|---|---|---|---|---|---|---|---|---|---|---|---|---|
| | | mAP$_b$ ↑ | AR$_n$ ↑ | WI↓ | AOSE↓ | mAP$_b$ ↑ | AR$_n$ ↑ | WI↓ | AOSE↓ | mAP$_b$ ↑ | AR$_n$ ↑ | WI↓ | AOSE↓ |
| OW-DETR Gupta et al. (2022) | inst-sem | 37.12 | 3.81 | 0.493 | 25 | 37.92 | 3.61 | 0.858 | 65 | 34.85 | 3.09 | 0.977 | 144 |
| PETS Liu et al. (2023) | | 38.11 | 4.17 | 0.525 | 24 | 38.90 | 3.73 | 0.884 | 66 | 37.04 | 3.31 | 1.011 | 151 |
| LODS Li et al. (2022) | | 38.27 | 3.81 | 0.506 | 22 | 39.75 | 3.61 | 0.950 | 63 | 38.99 | 2.99 | 1.102 | 143 |
| SF-OSDA Luo et al. (2023) | | 38.31 | 3.45 | 0.405 | 25 | 38.60 | 3.61 | 0.787 | 73 | 38.55 | 3.03 | 0.940 | 171 |
| GraphGen (Ours) | | **40.53** | **4.38** | **0.378** | **20** | **40.27** | **3.92** | **0.743** | **60** | **39.84** | **3.42** | **0.854** | **140** |
| OW-DETR Gupta et al. (2022) | hom-sem | 28.07 | 2.20 | 0.679 | 69 | 26.63 | 2.81 | 1.053 | 123 | 25.52 | 6.92 | 3.276 | 818 |
| PETS Liu et al. (2023) | | 28.30 | 2.90 | 1.138 | 53 | 27.39 | 3.76 | 1.962 | 116 | 25.91 | 7.64 | 5.234 | 811 |
| LODS Li et al. (2022) | | 28.85 | 4.19 | 1.498 | **48** | 28.35 | 5.08 | 2.297 | **75** | 27.21 | 8.49 | 4.571 | **75** |
| SF-OSDA Luo et al. (2023) | | 29.00 | 3.64 | 1.360 | 52 | 28.14 | 4.42 | 1.980 | 130 | 26.70 | 8.02 | 3.954 | 786 |
| GraphGen (Ours) | | **30.24** | **4.48** | **1.269** | 55 | **29.83** | **5.28** | **1.853** | 120 | **28.57** | **8.78** | **3.758** | 790 |
| OW-DETR Gupta et al. (2022) | freq-dec | 45.06 | 7.67 | 0.855 | 168 | 42.97 | 10.00 | 1.364 | 331 | 42.95 | 11.03 | 1.609 | 460 |
| PETS Liu et al. (2023) | | 45.95 | 8.27 | 0.889 | 173 | 44.69 | 10.34 | 1.362 | 355 | 43.63 | 11.30 | 1.749 | 505 |
| LODS Li et al. (2022) | | 46.13 | 8.63 | 1.090 | 216 | 45.03 | 11.01 | 1.651 | 409 | 44.80 | 11.91 | 1.943 | 577 |
| SF-OSDA Luo et al. (2023) | | 46.32 | 9.35 | 1.222 | 245 | 45.56 | 12.02 | 1.763 | 453 | 45.13 | 11.32 | 2.099 | 637 |
| GraphGen (Ours) | | **47.58** | **10.54** | **0.825** | **165** | **46.23** | **12.57** | **1.253** | **320** | **45.86** | **12.83** | **1.557** | **450** |
| OW-DETR Gupta et al. (2022) | freq-inc | 25.26 | 3.47 | 1.936 | 135 | 22.17 | 3.17 | 1.889 | 212 | 20.82 | 2.92 | 2.549 | 558 |
| PETS Liu et al. (2023) | | 25.40 | 3.84 | 1.812 | 113 | 22.25 | 3.28 | 2.165 | 191 | 21.46 | 2.98 | 2.700 | 528 |
| LODS Li et al. (2022) | | 25.92 | 3.58 | 3.094 | 109 | 23.69 | 3.46 | 2.841 | 181 | 21.66 | 3.07 | 3.197 | 515 |
| SF-OSDA Luo et al. (2023) | | 25.99 | 3.37 | 1.764 | 124 | 22.80 | 3.22 | 1.824 | 208 | 21.50 | 2.85 | 2.571 | 552 |
| GraphGen (Ours) | | **26.83** | **3.97** | **1.017** | **67** | **24.94** | **3.68** | **0.993** | **113** | **22.58** | **3.29** | **1.487** | **314** |

Here, $\mathcal{L}_{\text{det}}$ is the standard detection loss, $\mathcal{L}_{\text{graph-dom}}$ is the cyclic consistency loss for domain alignment (**Shift i**), $\mathcal{L}_{\text{graph-nov}}$ is the contrastive loss for novel class discovery (**Shift ii**), and $\mathcal{L}_{\text{graph-sf}}$ is the consistency loss for source-free adaptation (**Shift iii**). $\lambda_1$, $\lambda_2$, and $\lambda_3$ are balancing weights.

This integrated strategy ensures that progress in one area (e.g., domain alignment) benefits others (e.g., novelty discovery), creating synergy not possible with separate optimization. In practice, we first warm up with domain alignment, then gradually add novelty discovery and source-free consistency, letting the model build strong cross-domain correspondences before addressing the open-set challenge.

# 4 EXPERIMENTS

**Datasets and Evaluation Metrics** We evaluate GraphGen on two benchmarks: Cityscapes → Foggy Cityscapes and Pascal VOC → Clipart, representing domain shifts in weather and style. Novel class sets are defined by instance-level semantics ('inst-sem'), homonym-based semantics ('hom-sem'), frequency decrease ('freq-dec'), and frequency increase ('freq-inc'), each posing unique domain and class distribution challenges.

For each dataset pair, some categories are set as known (shared) and others as unknown (target-only), simulating the Category Gap. Performance is measured by Mean Average Precision (mAPb) for known class detection, Average Recall (ARn) for novel class discovery, Wilderness Impact (WI) for

Table 2: Quantitative results on source-free open-set object detection from PascalVOC to Clipart, with **Top-1** and Top-2 marked.

| Method | $\Omega_n$ | $\text{mAP}_b \uparrow$ | $\text{AR}_n \uparrow$ | WI$\downarrow$ | AOSE$\downarrow$ |
|---|---|---|---|---|---|
| OW-DETR Gupta et al. (2022) | | 15.56 | 30.50 | 8.028 | 3431 |
| PETS Liu et al. (2023) | | 16.05 | 25.38 | 9.085 | 4872 |
| LODS Li et al. (2022) | 6 | 16.65 | 21.44 | 9.413 | 5108 |
| SF-OSDA Luo et al. (2023) | | 16.67 | 31.93 | 7.927 | 4699 |
| GraphGen (ours) | | **17.20** | **32.15** | **7.654** | **3401** |
| OW-DETR Gupta et al. (2022) | | 14.85 | 31.85 | 8.467 | 3871 |
| PETS Liu et al. (2023) | | 15.98 | 25.71 | 9.674 | 5496 |
| LODS Li et al. (2022) | 8 | 15.69 | 22.25 | 9.966 | 5790 |
| SF-OSDA Luo et al. (2023) | | 16.16 | 32.38 | 8.289 | 5250 |
| GraphGen (ours) | | **16.88** | **33.12** | **8.001** | **3551** |
| OW-DETR Gupta et al. (2022) | | 14.27 | 34.06 | 9.866 | 4995 |
| PETS Liu et al. (2023) | | 14.37 | 26.89 | 10.752 | 6874 |
| LODS Li et al. (2022) | 10 | 14.12 | 22.42 | 10.538 | 7271 |
| SF-OSDA Luo et al. (2023) | | 15.22 | 34.77 | 9.314 | 6640 |
| GraphGen (ours) | | **15.88** | **35.77** | **9.114** | **4781** |

unknown object interference, and Area Open Set Error (AOSE) for known/unknown differentiation. We test scalability with 3, 4, and 5 novel classes.

**Implementation Details** We use a ResNet-50 backbone pretrained by self-supervised learning, optimized with Adam (lr 1e-4, batch size 4, weight decay 1e-4) on NVIDIA A100 GPUs. The model features cross-attention modules, adaptive feature alignment, and 5 warm-up epochs. Key parameters: loss weights $\lambda_1 = \lambda_2 = 0.1$, $\lambda_3 = 0.5$; similarity parameter $\gamma = 1.0$; novelty thresholds $\tau_{nov} = 0.5$, $\tau_{\text{known}} = 0.5$, $\tau_{\text{unknown}} = 0.5$; prototype momentum $\tau = 0.9$; teacher decay $\alpha = 0.9$; contrastive temperature $\tau_c = 0.1$, neighborhood $k = 5$. Both weak and strong augmentations are used. Hyperparameters are chosen on the labeled source-domain train/validation split (or taken from prior work) and then fixed for all target experiments; no labeled target data (validation or test) are used for tuning or model selection.

**Source-free Adaptation of Baselines** For non-source-free baselines such as OW-DETR Gupta et al. (2022), PETS Liu et al. (2023), LODS Li et al. (2022), and SF-OSDA Luo et al. (2023), we follow a unified protocol. Each detector is first trained on the labeled source domain with its recommended configuration; afterwards, all source images are discarded and never used during adaptation. On unlabeled target data, source-dependent components are replaced with operations on target pseudo-labels predicted by the source-trained detector (e.g., supervised detection losses on source images become losses on high-confidence target pseudo-labels, and source–target contrastive or adversarial terms are reformulated using differently augmented target batches only). All baselines and Graph-Gen share the same adaptation schedule, learning rate, and target-domain data augmentations. Table **??** in the appendix summarizes the source-free modifications for each baseline.

## 4.1 MAIN EMPIRICAL RESULTS

To demonstrate the superiority of GraphGen, we compare it with state-of-the-art methods, including OW-DETR Gupta et al. (2022), PETS Liu et al. (2023), LODS Li et al. (2022), and SF-OSDA Luo et al. (2023). All methods are re-implemented under source-free settings for fairness. We also unify the experimental environment and parameter settings for all methods.

Table 1 shows that GraphGen outperforms all other methods in open-set domain adaptation from Cityscapes to Foggy Cityscapes. In the most challenging 'freq-inc' setting with 5 novel classes, GraphGen achieves 22.58% $\text{mAP}_b$ and 3.29% $\text{AR}_n$, demonstrating strong robustness and generalization in complex scenarios. These results highlight the effectiveness of our approach, especially as the number of novel classes increases.

Table 2 further validates the robustness and adaptability of GraphGen on the PascalVOC to Clipart task without access to source data. With 8 novel target categories, GraphGen achieves 16.88% $\text{mAP}_b$

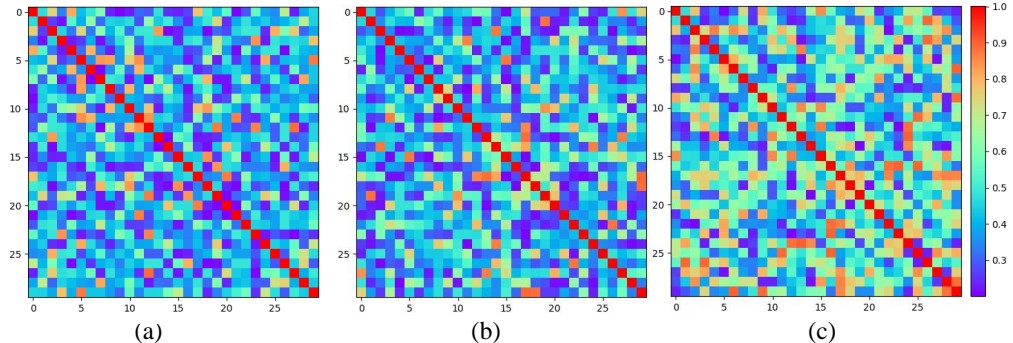

Figure 4: Box matching matrices between predicted and GT bounding boxes on the Cityscapes-to-Foggy Cityscapes domain shift. Matrices compare (a) DDETR, (b) OW-DETR, and (c) GraphGen . Strong diagonal elements indicate better matching performance.

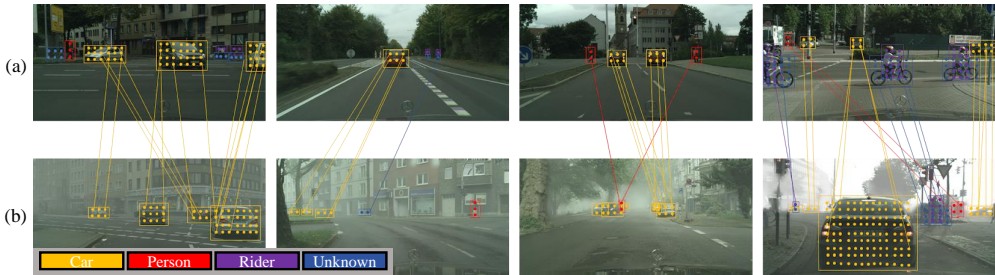

Figure 5: Post-hoc visualization of cross-domain object correspondence from Cityscapes to Foggy Cityscapes. (a) Source domain images (Cityscapes) and (b) target domain images (Foggy Cityscapes). Note: During adaptation, our method never accesses source images; this visualization is created after training solely to illustrate the learned cross-domain semantic alignment.

and 33.12% $AR_n$, setting new benchmarks for all source-free open-set metrics. The framework reduces WI (8.001) and AOSE (3551), indicating strong known-category performance. These gains mainly come from our dynamic graph architecture, which adaptively models instance relationships in the target domain.

Figure 3 presents qualitative comparisons among DDETR Wu et al. (2019), OW-DETR Gupta et al. (2022), and GraphGen on the Cityscapes to Foggy Cityscapes task. Green boxes indicate correct known detections, blue boxes represent correct unknown detections, and red boxes denote false positives. GraphGen better distinguishes between known and unknown objects under foggy conditions, reducing false positives. In contrast, baseline methods often suffer from overlapping detections.

## 4.2 FURTHER EMPIRICAL STUDY

**Ablation Study** We conduct ablation experiments to analyze the contribution of each Graph-Gen component. Tab. 3 shows the effect of our three main modules for the three shifts. The baseline without cross-domain graph alignment (No Graph-dom, for *Shift [i]*) achieves 37.86% $mAP_b$ and 2.11% $AR_n$. Adding graph alignment without cyclic consistency increases results to 38.22% $mAP_b$ and 3.33% $AR_n$. Removing graph-guided novelty discovery (No Graph-nov, for *Shift [ii]*) yields 38.55% $mAP_b$ and 3.03% $AR_n$. Without graph-aware self-regularization (No Graph-sf, for *Shift [iii]*), the model gets 39.96% $mAP_b$ and 3.33% $AR_n$. The full GraphGen combines all modules, achieving 41.06% $mAP_b$, 4.42% $AR_n$, and optimal WI/AOSE.

**Box Matching Analysis** Figure 4 shows box matching matrices between predicted and ground-truth boxes, comparing GraphGen with DDETR and OW-DETR. Each matrix shows the matching degree between predictions and ground truths across domains. Diagonal elements indicate strong correspondence and accurate detection. GraphGen shows stronger diagonals, meaning more precise and consistent box matching, while DDETR and OW-DETR have more off-diagonal elements, reflecting

Table 3: Ablation study on Source-free AOOD scenarios from Cityscapes to Foggy Cityscapes. The table shows the impact of different components with **Top-1** and Top-2 performance marked.

| Model Variants | $mAP_b \uparrow$ | $AR_n \uparrow$ | $WI \downarrow$ | $AOSE \downarrow$ |
|---|---|---|---|---|
| Source Only | 37.10 | 1.83 | 0.684 | 203 |
| No Graph-dom | 37.86 | 2.11 | 0.739 | 204 |
| Graph-dom w/o CC | 38.22 | 3.33 | 0.991 | 169 |
| No Graph-nov | 38.55 | 3.03 | 0.940 | 171 |
| No Graph-sf | 39.96 | 3.33 | **0.684** | **169** |
| GraphGen (Full) | **41.06** | **4.42** | **0.684** | **169** |

higher misalignment. This highlights GraphGen 's robust detection and matching across domains, even in open-set scenarios.

**Cross-domain Instance Matching** Figure 5 provides a visual representation of the cross-domain correspondence achieved by GraphGen using bipartite graph matching. In this figure, color-coded lines link corresponding objects between the source (clear) and target (foggy) domain images, highlighting the model's ability to maintain consistent semantic alignment across domains. The clear and foggy versions of the same scenes demonstrate how GraphGen accurately matches objects even under significant domain shifts, such as differences in visibility. This accurate cross-domain correspondence showcases the effectiveness of the bipartite graph mechanism in preserving object-level consistency and improving domain adaptation outcomes.

**Ablation Study** We conduct ablation experiments to assess each GraphGen component. Tab. 3 summarizes the effect of the three main modules. The baseline without cross-domain graph alignment (No Graph-dom) achieves 37.86% $mAP_b$ and 2.11% $AR_n$. Adding graph alignment without cyclic consistency improves results to 38.22% $mAP_b$ and 3.33% $AR_n$. Removing graph-guided novelty discovery (No Graph-nov) yields 38.55% $mAP_b$ and 3.03% $AR_n$. Without graph-aware self-regularization (No Graph-sf), the model gets 39.96% $mAP_b$ and 3.33% $AR_n$. The full GraphGen combines all modules, reaching 41.06% $mAP_b$, 4.42% $AR_n$, and optimal WI and AOSE.

**Computational Cost and Complexity** On Cityscapes→Foggy Cityscapes, GraphGen introduces a moderate overhead: confidence-based proposal filtering reduces nodes to roughly 260 per batch, and sparse k-NN adjacency (k=5) ensures O(Nk) rather than O(N²) complexity. The graph module adds about 12% latency and a small increase in memory while maintaining throughput in the same order as modern DAOD baselines, making it practical for real-world deployment.

**Multi-seed Robustness** To assess robustness, we also run three random seeds for the main Cityscapes→Foggy Cityscapes setting. GraphGen consistently outperforms the strongest source-free baseline with modest variance across seeds, indicating that our results are statistically stable.

**Hyperparameter Sensitivity and EMA Ablation** We further study the robustness of Graph-Gen to key hyperparameters, including novelty thresholds and EMA decay, on Cityscapes→Foggy Cityscapes. $mAP_b$ and $AR_n$ remain stable under $\pm 20\%$ perturbations of the default thresholds, and different EMA decays in the range $\alpha \in \{0.90, 0.95, 0.99\}$ yield similar final performance. A warm-up schedule (domain alignment first, then gradually adding novelty and self-regularization) leads to more stable training than enabling all modules from the start.

## 5 CONCLUSION

This paper introduces GraphGen, a unified framework addressing domain distribution shift, open-set class shift, and source-free transfer shift. Our approach leverages graph-based reasoning to model structural relationships between objects, enabling knowledge transfer without source data. Through graph-based feature alignment and novelty discovery, our framework effectively detects both known and unknown object categories in the target domain. Experiments demonstrate that GraphGen outperforms state-of-the-art methods in source-free open-set domain adaptation, with significant improvements in novel class discovery while maintaining strong performance on known categories.

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

## A  DECLARATION OF LLM USAGE

During the writing of the manuscript, we utilized a Large Language Model (ChatGPT) as a writing assistant. The scope of its usage was limited to **improving grammar, polishing sentences, and enhancing the clarity and fluency of this manuscript**. The method, claims, experimental results and conclusions are developed by the authors.

