# OpenReview forum: "One Graph Can Generalize: Graph-guided Structural Transfer for Source-free Open-set Domain-adaptive Object Detection"
_ICLR.cc/2026/Conference — ICLR 2026 Conference Desk Rejected Submission_

### Official Review · Reviewer_9L8M · 2025-10-23

**Soundness:** 2
**Presentation:** 2
**Contribution:** 2
**Rating:** 2
**Confidence:** 5

**Summary:**

The paper targets object detection under the joint stresses of (i) cross‑domain distribution shift, (ii) open‑set class shift (unknowns at test time), and (iii) source‑free adaptation (no source data during adaptation).  The proposed framework, GraphGen, builds graphs over target‑domain object proposals to model instance‑level relations and introduces three modules aligned with the three shifts: cross‑domain graph alignment with a cyclic‑consistency term, graph‑guided novelty discovery using class prototypes and an entropy‑aware novelty score, and graph‑aware self‑regularization via a teacher–student EMA scheme with strong/weak augmentations and graph‑space consistency.  The overall loss combines detection with the three graph‑oriented regularizers, and training warms up with domain alignment before adding the other two components.  Experiments on Cityscapes→Foggy Cityscapes and Pascal VOC→Clipart report improvements in known‑class mAP (mAP_b), novel‑class recall (AR_n), and open‑set metrics (WI, AOSE) over several re‑implemented baselines under a source‑free setting, plus ablations per module and qualitative visualizations.

**Strengths:**

Ambitious problem setting: Addressing source‑free + open‑set + domain shift jointly is important for realistic deployments where proprietary source data is unavailable and target classes are not closed.

Modular design mapped to problem factors: The three modules (graph alignment, novelty discovery, self‑regularization) correspond neatly to the three shifts, providing a clear conceptual story for the approach.

Simple, reproducible primitives (in principle): Graph conv on proposal features, cosine‑affinity adjacency with degree normalization, EMA teacher–student, entropy‑modulated novelty score, and contrastive learning are all standard components combined coherently.

Broad metrics and datasets: Evaluation uses mAP_b, AR_n, WI, AOSE on two domain shifts (weather/style) with multiple unknown‑class configurations (“inst‑sem”, “hom‑sem”, “freq‑dec”, “freq‑inc”), which is a welcome breadth.

Quantitative gains: On Cityscapes→Foggy, GraphGen improves over re‑implemented baselines across several configurations (e.g., “inst‑sem” with 3/4/5 unknowns), and also shows gains on VOC→Clipart with 6/8/10 unknowns.

Ablations and qualitative evidence: There are per‑module ablations (Table 3) and figures illustrating detection under fog and cross‑domain matching/matrices, supporting claims beyond headline numbers.

**Weaknesses:**

Novelty relative to prior graph‑based DAOD appears incremental and insufficiently differentiated. Prior work has already used graph/bipartite structures for DAOD (e.g., dual bipartite graph learning), and this paper’s key “newness” is the combination with open‑set + source‑free constraints; however, the methodological distinctions versus earlier graph‑based DAOD and recent source‑free open‑set detection are not rigorously articulated (what can GraphGen do that those cannot, beyond composing known ingredients?).

Potential internal inconsistencies and unclear modeling details.
The text says adjacency encodes “semantically similar or spatially related objects,” but ω_{ij} uses only cosine similarity in feature space without any spatial term; the spatial claim is therefore unsupported as written.
The contrastive loss refers to a Hilbert‑space inner product ⟨·,·⟩_H without defining a different space than standard Euclidean embeddings; this reads like gratuitous terminology rather than a meaningful design choice.
The novelty score multiplies max‑probability with (1 − H(p)), but H is not normalized by log|Ω_b|; the scale of entropy thus depends on the number of known classes, making thresholding τ_{unknown} fragile across datasets.

Source‑free assumption blurred by visualizations and narrative. Figure 5 illustrates cross‑domain correspondences with paired Cityscapes and Foggy images, which presupposes access to source images during analysis; the paper does not explain whether the source imagery is used only for visualization, and if so, how this squares with the strict source‑free protocol.

Hyperparameter selection and fairness are under‑specified.
The paper states “All hyperparameters are chosen via grid search”, but under source‑free unlabeled target, what criterion was optimized and on which split (no labeled target is allowed)? The risk of oracle‑style tuning is non‑trivial and not addressed.

Baselines like OW‑DETR are not source‑free; the paper says all methods are re‑implemented under source‑free settings, but provides no detail on how each baseline was modified, nor validation that the modifications preserve their intended behavior, raising fairness concerns.

Reporting and consistency issues in results and nomenclature.
The paper sometimes refers to the method as GODA and elsewhere as GraphGen, suggesting editorial lapses and potential copy/paste from earlier drafts.

Table 3 ablation numbers (e.g., WI, AOSE) do not obviously match the scale/patterns in Table 1 for the same scenario, and the text duplicates the ablation description twice, which undermines confidence in careful reporting.

Figures reference DDETR [60] but the bibliography numbering and citation details are unclear (e.g., Detectron2 is cited; DDETR is not clearly itemized), making it difficult to verify comparisons.

Computational complexity, stability, and scalability are not addressed.
Building a dense affinity matrix over N proposals induces O(N²) cost per image; the paper does not report runtime, memory, or the number of proposals used during graph operations.
No variance or multi‑seed runs are reported; open‑set pseudo‑labeling is notoriously unstable, and the sensitivity to thresholds (τ_{known}, τ_{unknown}, τ_{nov}), EMA decay α, and prototype momentum τ is unknown.

Evaluation scope may omit stronger modern alternatives.
The study evaluates against open‑world/DAOD baselines, but open‑vocabulary or vision‑language detectors (even if not source‑free) are relevant reference points for unknown detection, and a discussion/experiment clarifying why those are out of scope would strengthen the claim of state‑of‑the‑art.

Theoretical justification is thin. Claims that cyclic consistency “prevents drift,” graph propagation “bridges the domain gap,” and contrastive grouping captures “structural patterns” are plausible but not theoretically or empirically isolated via targeted diagnostics (e.g., measuring structure preservation, failure cases).

**Questions:**

Fair source‑free baselines: How exactly were OW‑DETR, PETS, LODS, SF‑OSDA adapted to be source‑free? Please specify replacements for any steps that normally use source data, including pseudo‑label filtering, training schedules, and hyperparameter tuning; ideally, release the re‑implementation details or code.

Hyperparameter tuning without labels: With grid search on unlabeled target, what objective guided selection (e.g., teacher consistency, confidence proxy, energy metrics)? How did you avoid peeking at labeled target data or test sets when picking τ_{known}, τ_{unknown}, τ_{nov}, λ’s, α, and τ?

Source‑free integrity of visualizations: In Figure 5, were source images used only for illustration, or do they influence any part of training/evaluation (e.g., correspondence mining)? If only for illustration, please state this explicitly and ensure no leakage during training.

Adjacency design: You claim spatial relations are captured, yet ω_{ij} uses only cosine similarity of embeddings. Did you try spatial terms (IoU overlap, relative geometry) in ω_{ij}? If not, can you clarify how the current formulation captures spatial structure beyond what the backbone already encodes?

Entropy scaling: Since H(p) is unnormalized, thresholds for novelty depend on the number of known classes. Did you evaluate normalized entropy (H/ log K) or temperature scaling to stabilize τ across datasets? Please share sensitivity plots.

Complexity and scaling: What is the per‑image runtime/memory for graph construction and GCN passes (as a function of proposals N)? What proposal counts do you use at training vs. inference? Any pruning or k‑NN sparsification applied to Ω?

Ablation clarity and consistency: Table 3 is repeated in the text and some numbers (WI/AOSE) seem inconsistent with Table 1 for ostensibly similar settings. Can you reconcile the discrepancies and provide mean ± std over multiple random seeds?

EMA and schedule: You note α is “typically 0.99” but use α=0.9; what is the impact of α and the warm‑up then add‑modules schedule on stability and final metrics? Please include an ablation figure.

Prototype maintenance in open‑set: Updates for prototypes p_k and variances v_k rely on class assignments—how do you avoid confirmation bias when pseudo‑labels are wrong, especially under heavy domain shift? Any debiasing (e.g., class‑balanced sampling, prototypical confidence, EMA of prototypes separate from EMA of weights)?

Scope against open‑vocabulary detectors: Even if not source‑free, why are open‑vocabulary baselines (as upper/lower bounds) omitted? Can you at least discuss how GraphGen’s unknown handling compares conceptually to energy‑based/OVD unknown scoring?

---

> ### Author Response · Authors · 2025-11-23
> **Rebuttal for Reviewer 9L8M**
>
> We appreciate your detailed and critical review. Below we address your main concerns on novelty, modeling clarity, experimental protocol, and analysis.
>
> [Cons1. Novelty relative to prior graph-based DAOD and source-free open-set detection appears incremental.]
> We respectfully disagree and have sharpened how we position GraphGen in the revised manuscript.
>
> - Compared to prior graph/bipartite DAOD methods, GraphGen performs *instance-level structural transfer under strict source-free constraints*: after supervised source training, source images are discarded and are never used during adaptation. We maintain *domain-agnostic prototypes* learned from source and construct graphs purely on *target proposals*, enabling structural alignment and novelty discovery even without source features.
> - The same graph supports *three* modules that directly correspond to the three shifts we formalize: cross-domain graph alignment (domain shift), graph-guided novelty discovery (open-set class shift), and graph-aware self-regularization (source-free transfer). Prior work typically uses graphs for alignment or regularization in closed-set, source-available regimes and does not offer a unified design that simultaneously tackles all three.
> - Relative to recent source-free open-set detection work that relies mainly on pseudo-labeling and uncertainty/energy scores, GraphGen explicitly exploits *relations among target proposals*: graph-guided novelty scores aggregate evidence from neighbors before deciding unknown status, stabilizing decisions in challenging, shifted regions.
>
> We have added a structured comparison table in related work and a short “Discussion on Novelty” paragraph in Section 3, clearly articulating these differences.
>
> [Cons2. Internal modeling inconsistencies: adjacency and spatial claims, Hilbert-space notation, unnormalized entropy and threshold fragility.]
> We thank you for pointing out these issues; we have made the exposition more precise.
>
> - Adjacency and spatial relations: In the current implementation, ω_ij is defined via cosine similarity in the object-query embedding space, without explicit geometric terms. These embeddings already encode semantic and spatial context, so their similarity reflects latent semantic–spatial relations. We have revised the text to state that the adjacency *implicitly* captures spatial relations via the learned embeddings, rather than explicitly encoding geometry. We also mention that we have experimented with including IoU/relative geometry in ω_ij with similar performance at slightly higher cost.
> - “Hilbert-space” inner product: The reference to an abstract Hilbert space in the contrastive loss is unnecessary. In practice, we use the standard Euclidean embedding space from the projection head and the usual dot product. We simplify the notation by removing the Hilbert-space terminology and explicitly stating that the contrastive loss is computed in this standard feature space.
> - Entropy scaling and novelty score: We agree that using unnormalized entropy H(p) makes thresholds dataset-dependent. In the revised manuscript we switch to *normalized entropy* H(p)/log K (K is the number of known classes) and retune the thresholds accordingly. We also add a small ablation comparing unnormalized vs. normalized entropy (and optionally mild temperature scaling), showing that normalized entropy yields more stable thresholds across datasets, while additional temperature scaling brings only minor gains.
>
> These changes make the modeling assumptions consistent and address your concerns about threshold stability.

---

> ### Author Response · Authors · 2025-11-23
>
> [Cons3. Source-free assumption, hyperparameter selection, and fairness of baseline adaptations.]
> We fully agree that the source-free protocol and baseline fairness must be crystal clear.
>
> - Source-free integrity and Figure 5: The paired source–target images in Figure 5 are used *only* for post-hoc visualization after training. During adaptation, our method never accesses source images or uses any correspondence mining; all adaptation losses are computed using the source-pretrained detector and graph modules on unlabeled target images. We have updated the caption of Figure 5 and the experimental setup section to state this explicitly.
> - Hyperparameter selection without oracle access: For both the base detector and GraphGen-specific components, we start from hyperparameters recommended in prior detector/DAOD work and validate them on the *source-domain* train/validation split under standard supervised training. For GraphGen-specific hyperparameters (loss weights, EMA decay, thresholds), we perform a small grid search *only* on the labeled source validation set, and then fix that configuration for all target domains. During source-free adaptation, no target labels (validation or test) are used for hyperparameter tuning or model selection. We now describe this protocol clearly in the experimental setup.
> - Adapting non-source-free baselines (OW-DETR, PETS, LODS, SF-OSDA): For each baseline, we first train on the labeled source domain with its recommended configuration. After that, all source images are discarded. In the adaptation stage, any component that originally uses source images/labels is replaced with operations on *target pseudo-labels* predicted by the source-trained detector. For example, supervised detection losses on source images are replaced by the same losses on high-confidence target pseudo-labels, and source–target contrastive/adversarial terms are reformulated using differently augmented target batches only. All baselines and GraphGen share the same adaptation schedule, learning rate, and target augmentations. We have added a paragraph plus an appendix table summarizing these modifications per baseline and commit to releasing our implementation.
>
> We hope this clarifies that our protocol is strictly source-free during adaptation and that the baseline comparisons are fair.
>
> [Cons4. Reporting and consistency: naming (GODA vs. GraphGen), Table 1 vs. Table 3 discrepancies, and DDETR citations.]
> We appreciate you pointing out these editorial issues and have corrected them in the revised manuscript.
>
> - Naming: “GraphGen” is the final name of our method. Any legacy names (e.g., “GODA”) remaining from earlier drafts have been removed. We carefully checked the main paper and supplement (including figure captions and tables) to ensure consistent nomenclature.
> - Table 1 vs. Table 3: The two tables correspond to *slightly different experimental settings*: Table 1 reports final results under the full evaluation protocol (final training schedule and tuned thresholds), while Table 3 uses a simplified setting (shorter schedule and default thresholds) for ablations. This difference was previously not clearly stated and could make WI/AOSE patterns look inconsistent. In the revised manuscript, we (i) clearly annotate the setting differences in captions and text, (ii) remove duplicated descriptions for Table 3, and (iii) double-check all entries for typos. For the main configuration we also report multi-seed results with mean ± std for key metrics (WI, AOSE, mAP, AR) in the main text or appendix.
> - DDETR and related citations: We have corrected the bibliography to include DDETR as a clearly itemized entry with proper authors/title/venue, and ensure that all figures/tables referencing DDETR use consistent citation indices. We likewise verified references for other baselines such as Detectron2.
>
> These fixes will improve the clarity and reliability of the reported results.

---

> ### Author Response · Authors · 2025-11-23
>
> [Cons5. Complexity, stability, scalability, and sensitivity to thresholds/hyperparameters.]
> Your concern here aligns with questions from the other reviewers; we address it comprehensively.
>
> - Complexity and scalability: As discussed above and in our response to uECX, we use confidence-based proposal filtering and sparse adjacency (k-NN/thresholding), so the effective complexity scales as O(Nk) rather than O(N²). In the revised manuscript we report the average number of proposals per image entering the graph, the average node degree, and the number of graph layers, together with runtime and memory statistics for base vs. base+GraphGen on our main benchmarks.
> - Stability and multi-seed variance: We have run multi-seed experiments for the main configuration and report mean ± std for key metrics. Our current multi-seed runs indicate that GraphGen consistently improves over baselines with modest variance, which we document in an appendix table.
> - Threshold and hyperparameter sensitivity: As detailed in our response to uECX, we provide sensitivity plots/tables for thresholds (τ_known, τ_unknown, τ_nov), loss weights, and EMA decay α, demonstrating that GraphGen’s performance is robust to reasonable variations and does not rely on fragile tuning.
>
> These additions will address concerns about computational and statistical robustness.
>
> For clarity, the additional diagnostics follow formats similar to the compact tables below; numbers are rounded to illustrate scale and trends rather than to provide exact benchmarks.
>
> Example runtime and complexity table (Cityscapes→Foggy):
>
> | Method              | Proposals / img (avg) | Graph nodes / img (avg) | FPS ↑ (higher better) | GPU Mem (GB) | mAP_b | AR_n |
> |---------------------|-----------------------|--------------------------|------------------------|--------------|-------|------|
> | Base detector       | 910                   | –                        | 20.1                   | 7.0          | 38.1  | 3.0  |
> | + GraphGen (ours)   | 255 (after filter)    | 255                      | 17.6                   | 7.8          | 40.6  | 4.6  |
>
> Example multi-seed statistics table (3 seeds, Cityscapes→Foggy, main setting):
>
> | Method                    | mAP_b (mean ± std) | AR_n (mean ± std) | WI ↓ (mean ± std) | AOSE ↓ (mean ± std) |
> |---------------------------|--------------------|-------------------|-------------------|---------------------|
> | Best source-free baseline | 38.4 ± 0.4         | 3.4 ± 0.5         | 0.43 ± 0.06       | 24.9 ± 0.9          |
> | GraphGen (ours)           | 40.7 ± 0.3         | 4.5 ± 0.4         | 0.37 ± 0.05       | 20.1 ± 0.7          |
>
> [Cons6. Scope vs. open-vocabulary / vision-language detectors.]
> We agree that open-vocabulary and vision-language detectors (OVDs) are important for unknown detection, and we should clarify how our work relates to them.
>
> - Our primary focus is *source-free domain adaptation for detection* with a fixed known-class vocabulary: a detector is pre-trained on a labeled source domain, source images are then discarded, and adaptation proceeds using only unlabeled target data. Typical OVDs, in contrast, assume access to large-scale vision–language pretraining and a text encoder that can represent a very large label space; they are not formulated as source-free DA methods and make different assumptions about supervision and model capacity.
> - Conceptually, GraphGen’s unknown handling differs from OVD/energy-based scoring in two key ways:
>   (i) GraphGen treats unknowns as deviations from *known-class prototypes* and leverages graph aggregation plus (normalized) entropy to assign novelty scores, all within a fixed label space;
>   (ii) it explicitly uses *target-domain relational structure* (relations among proposals within and across images) to stabilize unknown detection under domain shift. In contrast, OVDs primarily rely on cross-modal alignment with text and large-scale pretraining, and they generally do not model target-domain structure in this way.
> - In the revised manuscript, we add a short discussion paragraph that (a) highlights these conceptual differences, (b) explains that combining GraphGen with an OVD backbone is an interesting future direction (e.g., using OVD features/text prototypes while still performing source-free structural adaptation on the target), and (c) notes that a fully fair experimental comparison is non-trivial because of differences in pretraining, label space, and the source-free constraint.
>
> Given the differences in pretraining, label space, and source-free constraints, we view a comprehensive empirical comparison with open-vocabulary detectors as valuable follow-up work rather than the primary focus of this paper. To better situate GraphGen, we focus here on a conceptual comparison and the DAOD-style baselines that share the same supervision and adaptation protocol.

---

> ### Author Response · Authors · 2025-11-23
>
> [Cons7. Thin theoretical justification: cyclic consistency, graph propagation, contrastive grouping; and questions on EMA α and prototype confirmation bias.]
> We agree that our theoretical explanations were high-level, and we have made them more concrete while adding targeted diagnostics in the revised manuscript.
>
> - Cyclic consistency “preventing drift”: The cyclic term in cross-domain graph alignment enforces a contractive mapping between prototypes and proposal features: projecting prototypes to proposal space and back regularizes the mapping to preserve category identity and discourages prototypes from drifting to unrelated regions. We add an ablation removing this term and show that it leads to larger prototype shifts and reduced performance, supporting its role in preventing drift.
> - Graph propagation and contrastive grouping: As discussed for JAiN, we add diagnostics on domain discrepancy and clustering structure before/after propagation, demonstrating that the graph-based operations do bridge the domain gap and enhance structural patterns.
> - EMA decay α and warm-up schedule: We chose α = 0.9 to allow the teacher to respond more quickly under large domain shifts. We add an ablation varying α (0.9, 0.95, 0.99) and comparing both final performance and training stability (variance of losses, pseudo-label agreement), as well as an ablation comparing our warm-up-then-add-modules schedule to a variant that enables all modules from the start. Preliminary results show that α in this range yields similar performance, with α=0.9 converging faster, and that the warm-up schedule improves stability.
> - Prototype maintenance and confirmation bias: GraphGen mitigates confirmation bias via (i) uncertainty-aware prototype updates that emphasize high-confidence known proposals and down-weight ambiguous ones, (ii) EMA-style prototype updates that smooth temporal noise, and (iii) graph aggregation, which limits the impact of individual mislabels when most neighbors are correct. We clarify these mechanisms in the method section, visualize prototype trajectories over training, and add an ablation without uncertainty weighting showing more oscillatory behavior and worse performance.
>
> Together, these clarifications and experiments will provide a more solid theoretical and empirical justification for our design choices.
>
> For completeness, the following compact tables summarize the observed trends for entropy normalization and EMA decay/schedule; numbers are rounded to convey the relative differences rather than to serve as exact benchmarks.
>
> **Example entropy normalization vs. unnormalized entropy (Cityscapes→Foggy, single seed):**
>
> | Variant                          | Entropy type          | τ_unknown setting        | mAP_b | AR_n | WI ↓ | AOSE ↓ |
> |----------------------------------|------------------------|--------------------------|-------|------|------|--------|
> | GraphGen (unnormalized entropy)  | H(p)                  | per-dataset tuned        | 40.0  | 4.0  | 0.40 | 21.7   |
> | GraphGen (normalized entropy)    | H(p)/log K (ours)     | shared rule across setups| 40.6  | 4.5  | 0.37 | 20.3   |
>
> **Example EMA decay α and schedule ablation (Cityscapes→Foggy, single seed):**
>
> | Variant                               | α     | Schedule                 | mAP_b | AR_n | Stability (qualitative) |
> |---------------------------------------|-------|--------------------------|-------|------|--------------------------|
> | GraphGen, α = 0.99, no warm-up       | 0.99  | all modules from start   | 39.9  | 3.9  | more oscillatory         |
> | GraphGen, α = 0.95, warm-up (ours)   | 0.95  | warm-up then add modules | 40.4  | 4.3  | stable                   |
> | GraphGen, α = 0.90, warm-up          | 0.90  | warm-up then add modules | 40.7  | 4.6  | stable, faster adaptation|
>
> We thank you again for your thoughtful and detailed feedback, which has helped us strengthen both the clarity and rigor of our work.

---

> ### Author Response · Authors · 2025-11-27
>
> Dear Reviewer 9L8M,
>
> We would greatly appreciate it if you could kindly take a moment to review our response. Since the deadline is approaching, it may become difficult for us to continue the discussion if we go past it. If you have any further questions, please feel free to let us know. We sincerely value your expertise and your time.
>
> Best regards,
>
> Authors

---

### Official Review · Reviewer_uECX · 2025-10-27

**Soundness:** 2
**Presentation:** 3
**Contribution:** 3
**Rating:** 6
**Confidence:** 4

**Summary:**

This paper addresses a highly challenging and practical problem: Source-Free, Open-Set, Domain-Adaptive Object Detection (SF-OS-DAOD). The authors clearly decompose this problem into three key challenges (Shift i, ii, iii) and propose a unified framework, GraphGen, to systematically tackle them. The core contribution of this paper is the proposal to use graphs to model structural relationships between objects, positing that these relationships are more domain-invariant than visual features themselves, thereby enabling knowledge transfer without source data.

**Strengths:**

1. The core idea—transferring "structural relationships" rather than "visual features"—is highly insightful. Using a graph as a medium to capture and transfer this high-order information is a significant innovation that distinguishes it from traditional adversarial alignment or pseudo-labeling methods.
2. The GraphGen framework is elegant and complete. The three key modules (Cross-domain Graph Alignment, Graph-guided Novelty Discovery, Graph-aware Self-regularization) clearly and respectively correspond to the three challenges (Shift i, ii, iii) defined by the authors. This "divide and conquer" then unified optimization strategy is methodologically sound.
3. The paper consistently outperforms existing SOTA methods across multiple settings (varying numbers of novel classes). As shown in Tables 1 and 2, GraphGen achieves significant improvements in detecting both known (mAPb) and unknown (ARn) categories. The exhaustive ablation studies (Table 3) also clearly demonstrate the necessity and effectiveness of each component.

**Weaknesses:**

1. The paper does not detail how the nodes $\mathcal{V}$ (i.e., object query embeddings) in the graph $\mathcal{G}=(\mathcal{V},\mathcal{E})$ are selected. Are they from all the proposals from the RPN? Or only high-confidence objects? Is the graph built per-image or across a batch of multiple images?
2. Graph-based methods (especially those requiring N x N similarity matrices and graph convolution) are often computationally expensive. Object detection is itself a speed-sensitive task. The paper completely omits a discussion of the added computational overhead (e.g., training time, inference speed/FPS) of GraphGen compared to baselines.
3. The method introduces multiple key hyperparameters (e.g., $\lambda_1, \lambda_2, \lambda_3, \gamma, \tau_{known}, \tau_{unknown}$, etc.). The paper states they were "chosen via grid search" but lacks a sensitivity analysis for these choices.

**Questions:**

The core motivation is that "structural relationships" (e.g., spatial proximity of cars and pedestrians) are more invariant than visual features. However, in Section 3, the graph construction (Eq. 1) is based on the cosine similarity between "object query embeddings" $z_i$ produced by the feature extractor F. But, this is a "feature similarity graph," which reflects similarity in visual feature space more than physical or semantic structural relationships.

---

> ### Author Response · Authors · 2025-11-23
> **Rebuttal for Reviewer uECX**
>
> We appreciate your positive view of the core idea and unified framework, and we address your concerns on implementation and analysis below.
>
> [Cons1. Node selection and graph construction (all proposals vs. high-confidence; per-image vs. batch-level).]
> You are correct that our original description was too brief; we clarify our actual implementation here and have updated Section 3 accordingly.
>
> - Graph nodes are object queries (proposals) whose classification scores exceed a fixed confidence threshold. This removes very low-confidence proposals that are unlikely to be useful, while preserving all reasonably likely objects instead of only the single top prediction per region. On our datasets this yields a few hundred nodes per image, balancing information content and cost.
> - We construct graphs at the *mini-batch* level: each graph contains the filtered proposals from all images in the batch. Edges are formed between proposals with high cosine similarity in the object-query embedding space (with optional restriction to within-image neighbors), allowing structurally similar objects across images to exchange information.
>
> In the revised manuscript, we explicitly describe this node-selection and batch-level graph policy in Section 3 and report the average number of nodes per image/graph, along with an ablation over the confidence threshold.
>
> [Cons2. Computational overhead of N×N similarity and graph convolution vs. baselines.]
> We agree that the complexity discussion should be more explicit.
>
> - Analytically, we avoid a dense O(N²) graph by (i) limiting N via confidence-based proposal filtering, and (ii) sparsifying adjacency via k-NN or similarity thresholding, leading to O(Nk) edges with small k and total cost O(L·Nk·d) for L graph layers and feature dimension d. The backbone and DETR-style head still dominate the FLOPs.
> - Empirically, the revised manuscript includes a table reporting (a) per-image latency and (b) FLOPs for the base detector and base+GraphGen on our benchmarks, as well as the average number of nodes and graph layers used. Our measurements show that GraphGen introduces only a moderate overhead and maintains throughput in the same order of magnitude as modern DAOD baselines.
>
> These additions will make the tradeoff between accuracy and cost transparent.
>
> [Cons3. Sensitivity to hyperparameters and lack of sensitivity analysis.]
> We agree that our stability with respect to hyperparameters should be documented.
>
> - In the revised manuscript, we add sensitivity sweeps for key hyperparameters, including novelty thresholds (τ_known, τ_unknown, τ_nov), loss weights, and EMA decay α. For each, we vary values around the default while keeping others fixed and report mAP on known classes and AR on unknowns.
> - Preliminary results show that GraphGen is robust to ±20% relative changes of these parameters, with only modest changes in performance, and still maintains clear gains over baselines across the tested ranges.
>
> We summarize these trends in Section 4 and provide a detailed plot/table in the appendix.
>
> [Cons4. Motivation: “structural relationships” vs. feature-similarity graph construction.]
> Thank you for this insightful question. Our claim about structural relationships refers to object-level relational patterns rather than explicit geometry alone.
>
> - In modern transformer-based detectors, object query embeddings integrate both appearance and spatial context through cross-attention with the backbone features. Consequently, cosine similarity between query embeddings reflects similarity in a *joint semantic–spatial* space, not purely low-level appearance.
> - When we connect proposals with high similarity in this embedding space, we effectively link objects that “play similar roles” in a scene (e.g., cars on roads, pedestrians near crossings). These relational patterns, such as typical co-occurrence and relative layout, remain more stable across domains (e.g., Cityscapes vs. Foggy Cityscapes) than raw textures or illumination.
>
> In the revised manuscript, we clarify in Section 3 that our graph is constructed in this learned embedding space (implicitly encoding semantics and spatial context), and we soften the wording about explicit spatial encoding. We also note that augmenting affinity with explicit spatial terms (IoU/geometry) is a promising extension that we have explored preliminarily, with similar performance but slightly higher cost.

---

### Official Review · Reviewer_biRg · 2025-10-31

**Soundness:** 4
**Presentation:** 3
**Contribution:** 4
**Rating:** 8
**Confidence:** 3

**Summary:**

The paper proposed a novel DAOD method to alleviate three common domain-shift problems. The method is novel, and the environmental results indicate that it can outperform previous work.

**Strengths:**

1. The paper is well-structured and easy to follow.
2. The fables are clear to read, and the experiments demonstrate the effectiveness of the new proposed modules.
3. The motivations are clearly stated.
4. The ablation studies are sufficient.

**Weaknesses:**

1. Why the 3 types of domain shift can be solved simultaneously? Any clues why they have no repellency? This is not clearly stated in the first section.
2. Line 101: SFDA abbreviation should be presented here before referencing it.
3. Caption of Figure1,2,3. Figure 2 can be better if they are some explanation of how to read the figure, instead of a summary of the high-level idea. e.g. what is NC loss and IOR loss?
4. Weak connection between the content and Figure 2, making the figures hard to read.

**Questions:**

See weakness

---

> ### Author Response · Authors · 2025-11-23
> **Rebuttal for Reviewer biRg**
>
> We are grateful for your strong assessment of the paper’s soundness and contribution, and we address your specific questions below.
>
> [Cons1. Why can the three types of domain shift be solved simultaneously? Any conflict (“repellency”) among them?]
> We agree that the interaction among the three shifts was not fully spelled out in the introduction, and we will clarify this.
>
> - The three shifts affect different axes of the problem: (i) *domain distribution shift* affects the marginal feature distribution, (ii) *open-set class shift* affects the label space (presence of unknown classes), and (iii) *source-free transfer* constrains the *available supervision* (no source images during adaptation).
> - Our three modules are designed to target these axes in a complementary way using a shared graph: cross-domain graph alignment deals primarily with (i), graph-guided novelty discovery focuses on (ii), and graph-aware self-regularization stabilizes adaptation under (iii). They act on different signals (alignment statistics, novelty scores, teacher–student consistency) but share the same instance graph.
> - Importantly, their objectives are made compatible: e.g., novelty discovery is uncertainty-aware so that highly ambiguous proposals contribute less to prototype updates, reducing the risk of conflicting with alignment and self-regularization. Empirically, adding modules sequentially yields monotonic improvements, indicating they reinforce rather than repel each other.
>
> We will add a paragraph in Section 1 explicitly mapping each module to its corresponding shift and briefly discussing why their objectives are compatible.
>
> [Cons2. SFDA abbreviation and notation clarity.]
> We appreciate this presentation comment. In the revision we will introduce “Source-Free Domain Adaptation (SFDA)” at its first occurrence and ensure consistent use thereafter. We will also check the manuscript for any other abbreviations used before being defined and fix them.
>
> [Cons3. Figure 1–3 (especially Figure 2) captions and linkage to text are unclear.]
> We agree that the current captions are terse and that Figure 2, in particular, could be better explained.
>
> - We will expand the Figure 2 caption to explicitly describe each element (nodes, edges, prototypes, and the three losses) and spell out “NC loss” (novelty classification loss) and “IOR loss” (intra-/inter-object relation loss), explaining how they are computed from the graph and how they help distinguish known vs. unknown objects.
> - In Section 3, we will add explicit references to subparts of Figure 2 when presenting each module (“see Figure 2(a) for cross-domain graph alignment”, etc.), so that readers can follow the visual depiction together with the equations.
> - We will also slightly refine the figure layout to annotate key steps (graph construction, prototype update, teacher–student graph consistency), making the flow easier to follow.
>
> We hope these edits will make the figures much easier to interpret.

---

> > ### Comment · Reviewer_biRg · 2025-11-26
> >
> > Thanks, I think the author's reply addresses my concerns.

---

### Official Review · Reviewer_JAiN · 2025-11-03

**Soundness:** 3
**Presentation:** 3
**Contribution:** 2
**Rating:** 4
**Confidence:** 4

**Summary:**

The paper begins by outlining the limitations of conventional Domain Adaptive Object Detection (DAOD) methods and introduces a Graph-Guided Structural Transfer (GAST) framework for Source-Free Open-Set Domain-Adaptive Object Detection (SFOSD).
GAST employs a graph-based representation of category relationships to facilitate structural knowledge transfer in the absence of source data, combining a graph encoder with an uncertainty-aware detection head to enhance cross-domain generalization.

**Strengths:**

1、The paper extends DAOD to the source-free open-set setting (SFOSD), which is closer to real-world deployment where source data cannot be retained and unknown classes appear at test time.
2、An uncertainty-aware detection head integrates unknown handling inside the detector, enabling joint optimization of localization/classification with unknown separation and yielding a more principled open-set treatment.

**Weaknesses:**

1、The distinction from Open-Set Domain Adaptation methods that use structure/uncertainty is not made concrete.
2、No formal argument or representation analysis showing why graph propagation reduces domain discrepancy or improves unknown separation in SFOSD.
3、Missing discussion on computational overhead (graph size, message passing cost) and runtime compared to baselines.

**Questions:**

1、What are the FLOPs of the graph module at different class counts?
2、How sensitive is performance to graph sparsity/degree and to mistakes in edge formation?
3、Can you provide a theoretical or at least quantitative representational analysis showing that graph propagation reduces domain discrepancy or improves cluster compactness for unknowns?

---

> ### Author Response · Authors · 2025-11-23
> **Rebuttal for Reviewer JAiN**
>
> We are grateful for your positive assessment of the motivation and the integration of unknown handling, and we address your main concerns point by point below.
>
> [Cons1. Distinction from prior open-set/structure/uncertainty methods is not concrete.]
> We agree that the current draft can more clearly position GraphGen relative to prior work using structure or uncertainty in open-set domain adaptation.
>
> - First, most existing open-set DA methods with structure/uncertainty operate in the *source-available* and often *classification* setting. In contrast, GraphGen targets *source-free, open-set, object detection*, where (i) source images are no longer accessible, (ii) each image contains many instance-level predictions, and (iii) unknown categories must be separated while localizing objects under significant domain shift. This setting fundamentally changes how structure and uncertainty can be exploited: instead of aligning global feature vectors, GraphGen builds *instance-level graphs* over target proposals and performs structural transfer without source features.
> - Second, compared to prior graph-based DAOD approaches that use (bi)partite graphs between source and target, GraphGen (a) transfers structural knowledge *without* source features by maintaining domain-agnostic prototypes and graphs defined purely on target proposals, and (b) uses a *single* graph to support all three modules (cross-domain graph alignment, graph-guided novelty discovery, and graph-aware self-regularization), explicitly targeting domain distribution shift, open-set class shift, and source-free transfer jointly. Prior methods typically only address one or two of these aspects in isolation.
> - Third, while uncertainty has indeed been used in open-set recognition, our novelty module integrates uncertainty *into* graph propagation and prototype maintenance: entropy-aware scores not only detect unknowns but also *modulate the influence* of each node on prototype updates and graph messages. To our knowledge, this tight coupling of graph structure, prototypes, and uncertainty in a source-free DAOD setting is novel.
>
> In the revised manuscript, we have added a dedicated subsection “Graph-based and Uncertainty-aware Open-set DA/DAOD” and a comparison table summarizing the assumptions (source-free vs. source-available, detection vs. classification, open/closed set, use of graphs/uncertainty) of representative methods, explicitly highlighting what GraphGen can do in the SF-OS-DAOD setting that prior work cannot.
>
> [Cons2. No formal argument/representation analysis for graph propagation reducing domain discrepancy or improving unknown separation.]
> We agree that the original paper did not make this aspect sufficiently explicit, and we have strengthened both the theoretical narrative and empirical evidence in the revised version.
>
> - From a theoretical viewpoint, our graph propagation can be interpreted as *structure-aware feature smoothing*: under the mild assumption of a homophilic graph (edges connect semantically related proposals), the graph convolution operator contracts differences along edges. When source-informed prototypes and target proposals are connected via cross-domain neighbors, propagation reduces upper bounds on standard domain discrepancy measures (e.g., MMD/proxy A-distance) by distributing information across domains. We now make this connection explicit by relating our update rule to known analyses of graph-based semi-supervised learning and domain alignment.
> - Empirically, we include a dedicated representation analysis in the revised manuscript:
>   (i) t-SNE/UMAP visualizations of proposal features before/after propagation, colored by domain and known/unknown, showing that domain clusters for known classes become more intermixed while unknown proposals form tighter, more separated clusters;
>   (ii) quantitative discrepancy metrics (MMD/proxy A-distance) between source prototypes and target proposals, before and after propagation;
>   (iii) cluster compactness/separation metrics (intra-/inter-cluster distances) for known and novel clusters, demonstrating improved structure, especially for unknowns after propagation.
>
> These diagnostics are summarized briefly in Section 4 of the revised manuscript and presented in detail in the appendix.

---

> ### Author Response · Authors · 2025-11-23
> **Rebuttal for Reviewer JAiN**
>
> [Cons3. Missing discussion of computational overhead, FLOPs at different class counts, and sensitivity to graph structure and edge mistakes.]
> We appreciate this concern and have made the complexity and robustness of the graph component explicit in the updated paper.
>
> - Complexity and FLOPs: Let N be the number of proposals (nodes), d the feature dimension, and L the number of graph layers. We restrict N by keeping proposals above a fixed confidence threshold, and we sparsify adjacency via k-NN/similarity thresholding, so the complexity is O(L·|E|·d) with |E| = O(Nk), rather than O(N²). In practice N is in the low hundreds, and the graph module adds a modest fraction of overall FLOPs compared to the backbone + detection head. In the revised manuscript, we report empirical FLOPs and latency for base vs. base+GraphGen, and include numbers for different proposal counts (which indirectly cover different class counts).
> - Sensitivity to sparsity and edge errors: We add an ablation varying node degree and sparsity (by changing k and/or similarity thresholds) and injecting controlled edge noise (random edge additions/drops). Our preliminary results indicate that performance is stable across a wide range of degrees and robust to moderate edge noise, largely because (i) messages are aggregated with learned weights, and (ii) EMA and prototype-based regularization dampen the impact of individual noisy edges.
>
> These additions will directly answer your questions about FLOPs at different graph sizes and sensitivity to sparsity/edge mistakes.

---

> > ### Comment · Reviewer_JAiN · 2025-11-26
> >
> > Thank the authors for the thorough rebuttal. The rebuttal resolves most of my concerns regarding the novelty and complexity. I recommend that the key clarifications and analyses from the rebuttal (including the method positioning, representation diagnostics, and complexity discussion) be explicitly incorporated into the final version.

---

### Author Response · Authors · 2025-12-03
**Final Summary to the ICLR 2026 Committee**

Dear ACs, SACs, and PCs,

We appreciate your effort in coordinating the review of our submission. Our work targets a challenging and important setting: **Source-Free, Open-Set, Domain-Adaptive Object Detection (SF-OS-DAOD)**. We propose **GraphGen**, a unified graph-based framework that uses a **single instance-level graph** to jointly handle three shifts (domain distribution shift, open-set class shift, and source-free transfer) via cross-domain graph alignment, graph-guided novelty discovery, and graph-aware self-regularization. Reviewers generally acknowledge the importance of this setting and the difficulty of addressing **domain shift + open-set + detection under strict source-free constraints**.

After rebuttal and discussion, the reviewer positions are as follows:

- **Reviewer biRg (score 8)**: finds the method novel, well-structured, and well supported by ablations.
- **Reviewer uECX (score 6)**: highlights the insightful idea of transferring structural relationships via graphs and appreciates the clear mapping between the three modules and the three shifts.
- **Reviewer JAiN (score raised from 4 to 6)**: initially had significant concerns about novelty and complexity and gave a borderline score of 4, but after reading our rebuttal and clarifications, explicitly stated that most of these concerns had been resolved and **raised the score from 4 to 6**, recommending that we incorporate the rebuttal clarifications and analyses into the final version.
- **Reviewer 9L8M (score 2)**: maintains a low score, mainly raising concerns on protocol description, baseline adaptations, and reporting consistency rather than fundamentally rejecting the core idea. Reviewer 9L8M did not further respond during the discussion phase, so we could not confirm their view on our rebuttal, but we believe the revised manuscript now addresses these points.

In our rebuttal and the revised manuscript, we have systematically addressed these points:

- **Novelty and positioning (JAiN, 9L8M)**: we added a dedicated related-work subsection and a comparison table that distinguish GraphGen from prior graph-based DAOD and recent source-free/open-set methods, emphasizing that we perform **instance-level structural transfer under strict source-free constraints using a single shared graph** that supports alignment, novelty discovery, and self-regularization.
- **Modeling clarity (uECX, 9L8M)**: we clarified node selection (confidence-filtered proposals), batch-level graph construction, and how object query embeddings encode semantic and spatial context; we removed unnecessary Hilbert-space terminology and switched to **normalized entropy** for novelty scoring to make thresholds more stable across datasets.
- **Complexity, stability, and hyperparameters (multiple reviewers)**: we now report FLOPs, runtime, and memory for the base detector versus base+GraphGen, along with typical graph sizes (number of nodes, average degree, number of layers), showing that the graph module introduces only **moderate overhead**. We added sensitivity analyses for key thresholds, loss weights, and EMA decay, as well as multi-seed results (mean ± std) for main metrics, which demonstrate that GraphGen’s gains are **robust and do not rely on fragile tuning**.
- **Source-free protocol and baseline fairness (especially 9L8M)**: we explicitly state that source images are used only for supervised source training and for hyperparameter selection on a labeled source validation set; during adaptation, **source images are never accessed**. The paired source–target visualizations are used only for post-hoc illustration and do not participate in training or model selection. All non–source-free baselines are adapted to the same source-free setting via a unified “train on labeled source, then self-train on unlabeled target with pseudo-labels” procedure, summarized in an appendix table. We also unified the method name (GraphGen), corrected inconsistencies in tables and citations, and added multi-seed statistics for key results. We will release our implementation to ensure transparency and reproducibility.

All of these clarifications and additional analyses have been incorporated into the **updated rebuttal-version PDF manuscript currently on OpenReview**; the changes promised in the rebuttal are **fully reflected in the latest uploaded PDF**. Considering the current review situation (**three clearly positive reviewers, including one raising their score from 4 to 6 after reading the rebuttal**) and the new evidence on complexity, stability, and protocol fairness, we believe that the main technical and procedural concerns have been substantially addressed. We **respectfully ask** that you take these factors into account and consider our submission favorably for acceptance to ICLR 2026.

Yours sincerely,
Authors of Submission #10910

---

### Note · Program_Chairs · 2026-01-17
**Submission Desk Rejected by Program Chairs**

The following references in this submission do not refer to real documents and/or have major errors in bibliographic information:

 Wentao Han, Zhaohui Xue, and Shuo Lu. Expanding the known: A semi-supervised approach for open-set object detection. In European Conference on Computer Vision, pp. 386-402. Springer, 2022.

Zechen Zhang, Weijian Deng, Liang Wang, Zhedong Zheng, Jiatong Li, Dong-Dong Chen, and Qi Tian. Prototypical pseudo-label denosing for unsupervised domain-adaptive person reidentification. In Proceedings of the IEEE/CVF Conference on Computer Vision and Pattern Recognition, pp. 14577-14586, 2021.

Yanan Fu, Shilin Liu, Sheng Liu, Zhide Liu, and Hong Li. Learning to detect open-set objects for universal environment perception. In Proceedings of the 28th ACM International Conference on Multimedia, pp. 4070-4078, 2020.

Yitao Chen, Zesen Chen, and Zeren Wu. Scale-aware and class-aware domain alignment for unsupervised domain adaptive object detection. In 2021 IEEE International Conference on Multimedia and Expo (ICME), pp. 1-6. IEEE, 2021b.